# Shape Memory Polymer-Based Endovascular Devices: Design Criteria and Future Perspective

**DOI:** 10.3390/polym14132526

**Published:** 2022-06-21

**Authors:** Sergio A. Pineda-Castillo, Aryn M. Stiles, Bradley N. Bohnstedt, Hyowon Lee, Yingtao Liu, Chung-Hao Lee

**Affiliations:** 1Biomechanics and Biomaterials Design Laboratory (BBDL), The University of Oklahoma, Norman, OK 73019, USA; sergiopinedac@ou.edu (S.A.P.-C.); arynstiles@ou.edu (A.M.S.); 2Stephenson School of Biomedical Engineering, The University of Oklahoma, Norman, OK 73019, USA; 3School of Aerospace and Mechanical Engineering, The University of Oklahoma, Norman, OK 73019, USA; yingtao@ou.edu; 4Department of Neurological Surgery, Indiana University School of Medicine, Indianapolis, IN 46202, USA; bbohnste@iu.edu; 5Laboratory of Implantable Microsystems Research (LIMR), Weldon School of Biomedical Engineering, Birck Nanotechnology Center, Purdue University, West Lafayette, IN 47907, USA; hwlee@purdue.edu

**Keywords:** shape memory polymers, intracranial aneurysms, endovascular embolization, personalized aneurysm treatment, 3D printing

## Abstract

Devices for the endovascular embolization of intracranial aneurysms (ICAs) face limitations related to suboptimal rates of lasting complete occlusion. Incomplete occlusion frequently leads to residual flow within the aneurysm sac, which subsequently causes aneurysm recurrence needing surgical re-operation. An emerging method for improving the rates of complete occlusion both immediately after implant and in the longer run can be the fabrication of patient-specific materials for ICA embolization. Shape memory polymers (SMPs) are materials with great potential for this application, owing to their versatile and tunable shape memory properties that can be tailored to a patient’s aneurysm geometry and flow condition. In this review, we first present the state-of-the-art endovascular devices and their limitations in providing long-term complete occlusion. Then, we present methods for the fabrication of SMPs, the most prominent actuation methods for their shape recovery, and the potential of SMPs as endovascular devices for ICA embolization. Although SMPs are a promising alternative for the patient-specific treatment of ICAs, there are still limitations that need to be addressed for their application as an effective coil-free endovascular therapy.

## 1. Introduction

Intracranial aneurysms (ICAs) are the focal dilation of the wall of brain arteries. ICAs predominantly emerge at the Circle of Willis and are the main cause of subarachnoid hemorrhage—a condition that can lead to moderate-to-severe brain damage, long-term morbidity, and/or mortality [1,2,3]. Unruptured ICAs affect 3–5% of the general population. In particular, they are prevalent at ages of 40–65 years old [4] and most remain asymptomatic. However, 3–50 in every 100,000 ICAs will eventually rupture (causing subarachnoid hemorrhage), and the risk of this event is associated with multiple factors, including aneurysm size, location, sex, hypertension, smoking history, and people of Japanese or Finnish descent [4,5]. The rupture risk can potentially be mitigated by timely and prophylactic ICA therapeutics.

The current treatment of unruptured aneurysms involves microsurgical clip ligation or endovascular embolization. In microsurgical clipping, neurosurgeons first perform a craniotomy and then use a metallic clip to tighten the aneurysm neck that completely prevents the intra-aneurysmal flow (Figure 1, left). Due to its high invasiveness, microsurgical clipping poses higher procedural risks and may not be suitable for elder patients. On the other hand, endovascular embolization therapies serve as a minimally invasive alternative approach. In endovascular embolization, surgeons deliver platinum coils (or an embolic device) to the aneurysm sac that aims to minimize the intra-aneurysmal remanent flow (Figure 1, right). Overall, endovascular therapies have better clinical outcomes and are associated with reduced treatment-related mortality [6,7,8,9]. Despite this short-term therapeutic advantage, endovascular embolization therapies more frequently require a second procedure (i.e., re-operation) than microsurgical clipping, according to the randomized, multicenter phase III International Subarachnoid Aneurysm Trial (ISAT) [10]. The unsatisfactory long-term outcome of endovascular embolization is an emerging clinical issue that arises from aneurysm recurrence, where the aneurysm sac grows and the neck becomes incompletely occluded due to the gradual coil compaction over time and the resulting increased residual flow within the aneurysm sac [11].

In general, the endovascular embolization therapies are limited by their incapacity to completely occlude the complex 3D aneurysm shape. This incomplete occlusion has been characterized through the Raymond–Roy occlusion classification (RROC) scale: Class I (RROC-I) representing complete occlusion; Class II (RROC-II) for neck remnant; and Class III (RROC-III) for incomplete aneurysm occlusion. State-of-the-art embolic devices, such as Guglielmi detachable coils (GDCs), hydrogel-coated coils and the Woven EndoBridge, have an RROC-III rate in the range of 16–30% [12,13]. Therefore, there is an emerging need for the development of endovascular embolization devices that can provide a superior performance in achieving complete and *lasting* occlusion.

With this clinical need in mind, patient-specific considerations and personalized ICA treatment may serve as a promising approach to the development of such individual-optimized embolic devices. In this approach, patient cranial imaging data will first be used to reconstruct the 3D geometry of the aneurysm, and biomaterials can then be designed and fabricated for targeting patient-specific ICA and arterial geometries that can achieve complete and durable occlusion. In this setting, highly porous biomaterials will be an ideal candidate for such development, which can also aid in thrombogenesis—a fundamental process for preventing aneurysm recurrence. In this review article, we present the current literature of a specific class of biomaterials used in the design of endovascular devices: shape memory polymers (SMPs).

SMPs are the materials with the capacity to undergo thermo-mechanical changes in response to different stimuli. These polymer materials can be fabricated into a “home” configuration, where the material is rigid and not deformable. Then, by means of an external stimulus, the material can be deformed to a “stored” (also known as “reprogrammed”) configuration. With the second stimulus applied, the recovery of the material to their original geometry can be achieved. This unique feature poses great opportunities for designing responsive materials that enable personalized endovascular ICA treatment. For example, current endovascular devices, such as GDCs, can be improved by an SMP coating that occupies the gaps between coils (Figure 2a). Another potential approach can involve the fabrication of coil-free devices using patient-specific geometry to occlude the aneurysm after shape recovery (Figure 2b).

The remainder of this review is as follows. We will first discuss the state-of-the-art endovascular devices for ICA endovascular embolization, their mechanisms of action, and their advantages and limitations. Then, we will describe the status of SMP research: materials, polymeric architectures, chemical features, activation methods, and the recent efforts for the clinical translation of SMP-based endovascular devices toward ICA treatment. Finally, we will conclude this review with some remarks about future research and developments.

## 2. Current Endovascular Devices and Limitations

### 2.1. Generalities of Endovascular Therapies

Endovascular devices for ICA treatment were first developed by Guglielmi et al. [14,15,16] in 1990, where soft and thin platinum wires, called Guglielmi detachable coils (GDCs), were wrapped inside the aneurysm sac to prevent the intra-aneurysmal blood flow (Figure 3). In addition, GDCs were designed with the capacity to induce thrombogenesis, a process where platelets accumulate on the surface of the coil using electrical stimuli. These devices revolutionized the treatment of ICAs due to their minimally invasive approach using catheter-guided delivery. Ever since their first introduction, they have been the gold standard for the treatment of saccular ICAs.

GDC-based therapies have been further improved by the inclusion of assisting devices. For example, the treatment of wide-necked ICAs was not possible with GDCs alone, as coils might migrate into the bloodstream in the parent artery. This limitation was later overcome by integrating wire-meshed stents or balloons to mechanically support the implanted coils [17]. In addition, coatings for coils, such as hydrogel or porous polymers, have also been developed to increase the packing density—another challenge in conventional endovascular coil embolization (i.e., bare coils) [18,19]. In the following, we will present an overview of the most prominent developments of various endovascular devices for ICA embolization, as well as their limitations.

### 2.2. Hydrogel-Coated Coils

GDCs frequently face limitations in packing density due to the uneven spatial distribution of the coil inside the aneurysm [20]. To address this, coil coatings have been designed to reduce the gaps between coil wrappings. Hydrogel-coated coils are the most predominant designs for this approach (Figure 3). For example, the HydroCoil endovascular system has been extensively studied for its application in ICA treatment. This device aims to increase the volumetric filling of the aneurysm using a hydrogel, which swells in contact with liquids. The HydroCoil endovascular aneurysm occlusion and packing study (HELPS) compared their performance to bare GDCs, where they found that the major aneurysm recurrence was reduced with the use of the hydrogel-coated coil, but other clinical outcomes had a similar performance to the control group (i.e., bare GDCs) [21,22]. This led to the subsequent development of the HydroSoft, a more advanced hydrogel-based coil, where the hydrogel was used as the inner core of the device. The HydroSoft aimed to reduce the limitations of the HydroCoil that introduced coil stiffness and a highly sensitive time restriction for its delivery. Guo et al. compared the use of these two devices and found that the HydroSoft coils had lower rates of neck remnant (RROC-II) than the HydroCoil [23]. The German–French Randomized Endovascular Aneurysm Trial (GREAT) also tested the HydroSoft device and compared it to bare GDCs [13]. This multicenter study found that composite angiographic and clinical outcomes were improved in the HydroSoft group, while adverse and serious adverse events occurred at similar rates in both groups. However, other clinical trials, including the Patients Prone to Recurrence After Endovascular Treatment (PRET) trial, have demonstrated that hydrogel-coated coils do not provide improved clinical outcomes when compared to bare coils [24]. Overall, the meta-analysis of the performance of hydrogel coils demonstrated that the second-generation hydrogel coils (HydroSoft) might improve residual aneurysm and midterm recurrence, but do not exhibit any other significant differences to the traditional GDC coiling [25].

### 2.3. Hydrogel/Liquid Embolic Materials

As another ICA treatment technique, coil-free materials were developed to occlude ICAs and, potentially, result in a higher rate of RROC-I. Materials with induced polymerization were used for these applications (Figure 3). For example, Poupart et al. designed a liquid polyethylene glycol dimethacrylate (PEGDMA) precursor with the potential to be delivered to an in-vitro aneurysm model and to be photopolymerized using ultraviolet (UV) light in situ. This material exhibits great potential for its in-vitro application, due to its ease-of-use and low erosion rates [26,27]. However, in-vivo testing/evaluation is yet to be performed to understand the complete potential of this device.

Another liquid embolic material with great potential for translation is Onyx. This material is a copolymer of ethylene and vinyl alcohol dissolved in dimethyl sulfoxide (DMSO). When in contact with other liquids, DMSO rapidly phase separates from the co-polymer and induces in-situ solidification. This mechanism allows for the delivery of the embolic material at the aneurysm lumen. The early in-vivo testing of the Onyx device in swine models demonstrated that occlusion with the material alone was effective, although with fairly long delivery times (∼40 min) and with cases of migration into the parent artery [28]. Onyx has been translated into clinical use, where excellent occlusion rates have been reported [29]. Nonetheless, complications of mass effect, DMSO toxicity, and off-target embolization have been reported [30].

### 2.4. Woven EndoBridge (WEB)

The treatment of wide-necked bifurcated intracranial aneurysms (WNBIAs) can be difficult if performed with traditional endovascular devices. The WEB device is a nitinol mesh oblate spheroid device that is expanded inside the aneurysm space and is specifically designed for WNBIAs (Figure 3) [31]. Several clinical studies were performed for testing the WEB device in ruptured and unruptured bifurcation ICAs [32,33,34,35], where it has been demonstrated that the WEB can provide the technically straightforward occlusion of ICAs. Furthermore, the WEB provides the great advantage of not requiring anti-platelet therapy post-implantation, while maintaining acceptable rates of immediate occlusion (∼80%) [12,36]. Overall, the WEB provides great improvement in the treatment of WNBIAs, making it a great candidate for its expansion into other types of ICAs. However, aneurysm recurrence is still a challenge and re-treatment is oftentimes required.

### 2.5. Flow Diversion Devices (Flow Diverters)

Flow diversion is a technique that aims at preventing intra-aneurysmal flow using a fine mesh that resembles a stent (Figure 3). This mechanism is thought to induce the formation of a neoendothelium and, ultimately, completely stop the blood flow into the aneurysm space [37]. In addition, flow diverters are targeted for the treatment of side-walled aneurysms, where the use of other embolic devices can be technically difficult. The first flow diverter device, named the Pipeline Embolization Device, was introduced in 2008. Its current iteration—the Pipeline Flex—recently received FDA approval in 2021 with improved surface technology for the prevention of thrombogenesis. The outcomes of this device, and other versions of flow diverters, have demonstrated its safe use [38]. Complete aneurysm occlusion has been achieved in ∼80% of patients treated with this device [38,39,40], and similar rates have been reported for other flow diversion devices, such as the Flow-Redirection Endoluminal Device (FRED) (82.5%) [41], the Tubridge (75.3%) [42,43], and the p64 flow modulation device (80.2%) [44]. However, flow diverters cannot be used for other aneurysm geometries. Furthermore, the mechanism for ICA embolization is not immediate, as it relies on the progressive clotting of blood within the aneurysm. For example, the FRED device exhibits the complete occlusion rates of 20% at 90 days of follow up, which increased to 82.5% at 180 days, 91.3% at 1 year, and 95.3% >1 year post implant. This implies that the aneurysm sac will be subjected to residual flow for at least a year after treatment, increasing its risk of rupture.

### 2.6. TrelliX

TrelliX is another alternative to coated coils, which has a similar approach to the hydrogel coils. Trellix coils are coated with a polyurethane (PU) SMP foam, which allows the device to occupy 2.5 times more volume than bare GDCs after its activation [45] (Figure 2a). The expansion of the foam can be triggered after aneurysm coiling, which is hypothesized to increase packing density and prevent aneurysm recurrence. In addition, the polymer formulation of this device has been reported to promote aneurysm healing through neoendothelium formation and increased collagen deposition on the surface of the material [19,46,47] (Figure 4). These devices have also been reported to have acceptable occlusion rates in in-vivo rabbit animal models, where progressive occlusion was observed [19]. However, its RROC occlusion rates were not systemically evaluated and reported. In addition, TrelliX is currently being tested in clinical trials (NCT03988062), where the long-term occlusion rates in humans are currently being evaluated and will be reported in July 2022.

### 2.7. Summary of Current Endovascular Devices

The state-of-the-art technology for the treatment of ICAs has allowed to expand the range of aneurysm types that can be treated with endovascular approaches. However, endovascular devices still face limitations related to incomplete occlusion, mass effect, devices migration, among others. Table 1 summarizes the previously described advantages and limitations of the endovascular devices.

## 3. Shape Memory Polymers for Endovascular Embolization of ICAs

The unique characteristic of SMPs undergoing thermo-mechanical changes as a response to stimuli poses a great opportunity for endovascular ICA embolization therapy. In theory, SMPs could be transported to the aneurysm space in a “stored” compressed geometry and, after stimulation, triggered to recover a given configuration and occlude the aneurysm. In this section, we will describe the unique properties of SMPs, the design and manufacturing techniques of these materials, different activation mechanisms for shape recovery, and some key design criteria for its application as a potential patient-specific endovascular embolization device.

### 3.1. Polymer Architecture

SMPs can be fabricated with different architectures that govern their shape recovery behavior. Specifically, the type of crosslinking between polymeric chains will aid in the fine-tuning of their transition temperature ($T_{trans}$) [48].

#### 3.1.1. Physically Crosslinked Co-Polymers

Physically crosslinked SMPs are linear co-polymers that consist of soft and hard segments. In this type of SMP, a deformable/rubbery state is achieved by increasing the material’s temperature (*T*) above its glass transition temperature ($T_{g}$) (i.e., $T_{trans}$ = $T_{g}$). When T< $T_{g}$, the permanent/glassy state of the material is achieved through weak interactions between the hard segments (e.g., van der Waals forces, dipole–dipole interactions or hydrogen bonding) [48,49]. The most common example of a physically crosslinked SMP is amorphous polyurethane, which is generally synthesized through the polymerization of diisocyanates as hard segments, and polyols as soft segments. Other examples include polyesterurethanes and polyetheresterurethanes [48,50,51].

#### 3.1.2. Covalently Crosslinked Copolymers

The covalent crosslinking of SMPs can be achieved through the use of monomers that induce the crosslinking of elongated chains or by post-processing [48]. The crosslinking can provide additional phases to the material, as well as the control over mechanical properties by altering the crosslink degree. Due to the rigidity of these bonds, covalently crosslinked SMPs cannot be extruded as traditional 3D printing filaments, limiting their processing into specific geometries [52]. Some examples of this type of SMPs include glassy thermosets (e.g., as epoxy SMPs [53]), and semicrystalline rubbers such as polyethylene [49,54].

### 3.2. Porous Shape Memory Polymers

Traditional and coated GDCs, as well as the WEB device, are transported through a microcatheter and are coiled within the aneurysm. Currently, SMP-based endovascular devices are under development so that they can be utilized in a similar manner. SMP design for endovascular devices can follow two major approaches: (i) SMP-coated coils such as Trellix [19,46]; and (ii) coil-free SMPs foams [55,56,57,58]. SMP-coated coils have a similar goal to the hydrogel coils: increasing the packing density may augment the volume of the coil. On the other hand, coil-free SMPs aim to be fabricated with patient-specific aneurysm geometry to obtain complete occlusion. Both approaches require the high compressibility of the material, which can be achieved through pore formation. In this section, we will review some existing methods for the fabrication of highly porous SMPs.

#### 3.2.1. Gas Foaming

Gas foaming is a process wherein the decomposition of a compound, known as a blowing agent, yields the production of gas and the formation of bubbles in the polymeric solution [59] (Figure 5). Other types of blowing agents include physical and biological agents. Physical foam formation can be realized through shaking, whipping, and other mechanical methods. In contrast, biological agents frequently rely on gas-producing species such as yeast [60]. For the fabrication of highly porous foams, excess isocyanate solutions in conjunction with commercial blowing agents can be utilized for polyurethane SMPs. For example, Singhal et al. used Enovate, a hydrofluorocarbon, for the fabrication of porous SMPs [61]. This blowing agent has been extensively applied for foam creation, although environmental agencies have recommended against its use [62,63]. Other methods applicable to SMP foam fabrication include CO${}_{2}$ [64], deionized water [65], and microwaving [66], to name a few.

#### 3.2.2. Particle Leaching

Solid particle leaching uses water-soluble particles for the fabrication of pores in SMPs. Leaching agents, typically particles of a given grain size and geometry, are dispersed in polymeric solutions before the curing of the SMP. After curing, particles are dissolved from the matrix by a solvent, leaving gaps in the polymeric structure (Figure 5). This method provides the high tunability of the cell/pore shape and size, which depend on the choice of particles. Common leaching agents that have been used in the literature include sodium chloride [67,68], sugar [56,58], and gelatin [69], among others. In our group, we leached polyurethane SMPs using sugar particles and demonstrated its use to obtain highly compressible SMP foams with pore geometries that mimic the sugar crystal morphology [56,58].

#### 3.2.3. Additive Manufacturing (3D Printing)

The 3D printing of materials can be performed via different methods, where fused deposition modeling (FDM) is the most popular. Three-dimensional printing provides a great opportunity to fabricate patient-specific geometries for SMP-based endovascular devices (Figure 5). By means of obtaining the aneurysm geometry through patient imaging data (e.g., magnetic resonance or computed-tomography imaging), a patient-specific aneurysm geometry could be printed using FDM to obtain a porous endovascular device. FDM has been utilized for the printing of materials with shape memory capacities, including the use of polylactic acid (PLA) [70,71,72], polyurethane [73,74,75], polycyclooctene (PCO) [76], polyvinyl alcohol (PVA) [77], among others [78].

For example, Langford et al. used PLA to design a highly compressible scaffold with shape memory properties. They developed a 3D geometry based on origami tessellations, allowing the fine control of the direction of compression and the shape recovery of the material. In addition, they designed the core of the material to have trabeculae and mimic the bone macrostructure, which can be used for bone tissue engineering [70]. A different approach for guided compression was developed by Mehrpouya et al., where a honeycomb structure was used to print PLA-guided SMPs [71]. These designs can also be a promising approach for the design of personalized endovascular devices tailored to patient-specific ICA geometries. Other examples for 3D-printed SMPs include stents [72] and wearable devices [74].

### 3.3. Actuators for SMPs

The shape recovery of SMPs can be triggered by different mechanisms, including pH [79], humidity [80], light triggering [81], magnetic activation [82], and resistive heating (i.e., Joule heating) [83]. The choice of actuation mechanism for shape recovery is a fundamental design criterion for the development of endovascular devices. In theory, when an SMP-based device is transported via the microcatheter to the aneurysm sac, it should be able to maintain its compressed/deformed configuration until the shape recovery has been triggered by the activation mechanism. In this subsection, we will present some of the most commonly used methods for shape recovery actuation.

#### 3.3.1. Joule Heating Activation

Joule heating is a process by which a material emits heat when electrical current flows through it. Therefore, this mechanism can provide a highly controllable trigger for shape recovery in SMPs. However, most SMP materials are not conductive, which creates the necessity of additives that induce conductivity. The current developments for electrically triggered SMPs include the use of carbon nanotubes (CNTs), carbon black, polypyrrole (PPy) coating, and nanoparticles [84].

In our group, we developed conductive SMPs via CNT infiltration in sugar-leached polyurethane foams [56,58]. We demonstrated that CNTs are an effective and facile method for the shape recovery of the porous SMP foams. SMP/CNT foams emit heat when the direct current is applied, allowing the recovery of the compressed foam (Figure 6). However, we also observed that CNT infiltration reduced $T_{g}$ of the material and deteriorated its capacity to withstand cyclic deformation [58]. Other researchers have also found similar behaviors for CNT-infiltrated SMP composites [85,86].

On the other hand, PPy is another alternative for the induction of conductivity. Typically, PPy is used as a coating of the SMP materials and its polymerization is performed in-situ. For example, Sahoo et al. immersed their SMP films in a pyrrole aqueous solution and then transferred them to a separate FeCl${}_{3}$ solution. This produced a thin layer of PPy on the surface of the SMP films and provided the material the capacity to shape recovery under electrical stimulation. As the accompanied by-product, PPy coating induced an increased $T_{g}$ [83]. Additionally, modifications to the PPy polymerization have also been studied. For example, Zhang et al. immersed PLA electrospun SMPs in an FeCl${}_{3}$ solution and then evaporated pyrrole onto the surface of the material. This coating induced a slight decrease in $T_{g}$ and minimal changes in the material’s mechanical properties (e.g., Young’s modulus and ultimate tensile strength) [51]. Similar results have been observed in other studies using PPy coating [87,88].

Overall, material modification for the induction of conductivity needs careful design. While conductivity can be achieved through the use of additives and coatings, the thermo-mechanical properties of the SMPs might also be undesirably affected by these processes.

#### 3.3.2. Infrared (IR) Light Activation

Infrared light can be absorbed by SMP materials as will then be transformed into heat. This activation method avoids the infiltration of the material with other additives and prevents the involuntary stimulation of the vascular tissue, due to the low IR absorbance of soft tissues. Leng et al. induced the recovery of a polystyrene/polyvinyl SMP using an IR laser, although their material did not fully recover in the whole process due to the slow temperature gain of the system [89]. A similar problem was observed by Baer et al.; they used a polyurethane SMP to fabricate a vascular stent with photothermal actuation [81]. The shape recovery of the stent was triggered within a mock artery model with active flow using a 810 nm diode laser. They observed that the shape recovery of SMP was successfully triggered in the artery phantom (without flow) and complete recovery was achieved in ∼7 min (that is much longer than the GDC delivery time: 1–2 min). However, when flow was applied to the artery phantom, the convective cooling prevented the triggering of the stent [81]. Overall, IR photothermal activation has great potential for its application in SMP endovascular devices, but further research is required to improve the heating rates.

#### 3.3.3. Electromagnetic Activation

The activation of shape recovery through remote signals can aid in the ease of the delivery of endovascular devices. Based on this, several efforts have been made to fabricate SMPs sensitive to electromagnetic waves. For example, Schmidt fabricated biodegradable SMPs using oligo(e-caprolactone)dimethacrylates and butylacrylate (BA), which were infiltrated with Fe${}_{3}$O${}_{4}$ nanoparticles (d≈11 nm). These nanoparticles can transform electromagnetic radiation (radio waves of λ=300 kHz, power of 5.0 kW) into the heat that triggered the shape recovery of the SMP [90]. Similar designs have been made with carbon black, nickel [86], and CNTs [91].

#### 3.3.4. pH Activation

pH-activated SMPs have been developed to prevent the heating of SMP devices or elements in the bloodstream of the parent artery. Using the well-described pH of blood, SMPs can be activated when in the presence of the fluid. Han et al. developed a β-cyclodextrin-modified alginate/diethylenetriamine-modified alginate (β-CDAlg/DETA-Alg) SMP with high deformability at pH =7. They demonstrated that the material conserved the deformed configuration when the aqueous solution pH was changed to 11.5. Then, shape recovery was triggered by transferring the material to the original solution [79]. Another use of alginate-based pH-activated SMPs was studied by Meng et al., where they fabricated a phenylboronic-acid-grafted alginate/PVA hydrogel. The material was deformed and activated and deformed at pH 6 and was also sensitive to glucose [92]. Another approach for pH-induced behaviors was demonstrated by Lu et al., where they fabricated a pH-activated SMP with self-healing capacity using four-armed poly(ethylene glycol) (PEG) with dopamine end groups loaded with Fe${}^{3+}$ ions. This material uses pH sensitive reversible mono-, bis- and tris-catechol–Fe${}^{3+}$ coordination as a switch for shape recovery and self-healing. By setting the solution pH to 9, the material exhibited excellent shape recovery behaviors and self-healing after an induced wound [93]. Similar behaviors and mechanisms were developed for water-activated SMPs [94,95]. pH-activated SMPs have great potential to be used as endovascular devices due to the absence of heating elements for shape recovery, making devices safer and simple for delivery.

Overall, the research of SMP is an evolving field with in which novel materials and synthesis methods are constantly emerging. For the most current developments, we present a summary of all the materials and activation methods pertinent to this review in Table 2.

## 4. Potential of SMPs for ICA Endovascular Therapies and Future Directions

Current developments have allowed the expansion of endovascular therapy to a wider spectrum of aneurysm morphologies, including giant, wide-necked, and fusiform aneurysms. However, most endovascular devices still face limitations with suboptimal outcome (i.e., high recurrence rates). As discussed in Section 2, the complete occlusion of aneurysms is only achieved in ∼80% of the treated cases. This incomplete occlusion allows residual flow to further promote aneurysm recurrence [116,117]. In fact, several coated coil devices have been designed to increase the packing density. However, it has been demonstrated that the most influential risk factor for aneurysm recurrence is the residual flow, rather than the packing density [11]. Therefore, it is of paramount importance that the novel developments of endovascular therapies aim to reduce residual flow while increasing complete occlusion rates (i.e., much more treated cases with RROC-I score). The fabrication of a porous patient-specific SMP is a very promising alternative for the next generation of endovascular devices. In theory, aneurysm-specific SMPs could completely occlude the aneurysm space and promote vascular healing. The latter can be induced by the biocompatibility and cellular adherence properties of the material. Most importantly, aneurysm healing can also be improved through advanced surface modification.

We also envision that the next generation of SMP-based endovascular devices will need to address several limitations of these materials before they can be translated into the clinic. For example, SMPs are very sensitive to environmental conditions and the shape recovery properties can be easily affected by environmental humidity, oxidation, and heat [100,101,118,119,120]. This implies that the storage conditions for SMPs need to be carefully investigated. In addition, the storage of the material also needs to guarantee sustained sterility. SMP sterilization is a controversial topic, as methods that are applied to other medical devices might not be readily suitable for this type of material. For example, the autoclaving or plasma sterilization of polyurethane SMPs can be detrimental to its mechanical properties and shape memory/recovery features by significantly changing $T_{g}$ [121,122]. Last but not least, the activation methods for SMP recovery are still a matter of debate, due to the potential use of heat sources for shape recovery. Alternatives including pH activation will aid in the advancement of safe shape recovery triggering.

## 5. Conclusions

In conclusion, SMPs are unique materials with a great potential in the medical field and in many other industries. Their development for endovascular embolization is promising and multiple efforts are being made in academia and in the industry for their translation. For ICA therapeutics, SMPs can improve the current approaches for aneurysm embolization by reducing the residual flow and preventing aneurysm recurrence. This can be made possible through the personalization of endovascular devices with aneurysm-specific geometries. This is a novel research idea that will need to be tested using in-vitro (via aneurysm phantoms and flow loops) and in-vivo (via the use of animal models [123]) techniques prior to their application in humans.

In this review, we discussed state-of-the-art devices for endovascular embolization and their limitations and presented a great alternative for the development of technologies with improved long-term outcomes. Aneurysm recurrence is attributed to the residual blood flow in aneurysms with incomplete occlusion. Coil-free SMP technologies have a great potential to address this extant limitation of existing endovascular ICA embolization devices. By addressing the current limitations of SMP materials presented in this review, endovascular embolization will see the appearance of the next generation of devices that can offer durable and personalized ICA therapy.

## Figures and Tables

**Figure 1 polymers-14-02526-f001:**
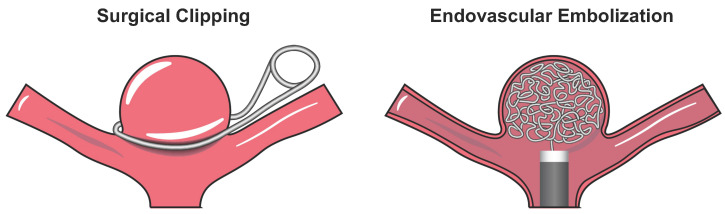
Schematic of the two primary aneurysm treatment methods: (i) surgical clipping (the metal clip not drawn to scale) (**left**); and (ii) endovascular embolization using Guglielmi detachable coils (GDCs) (**right**).

**Figure 2 polymers-14-02526-f002:**
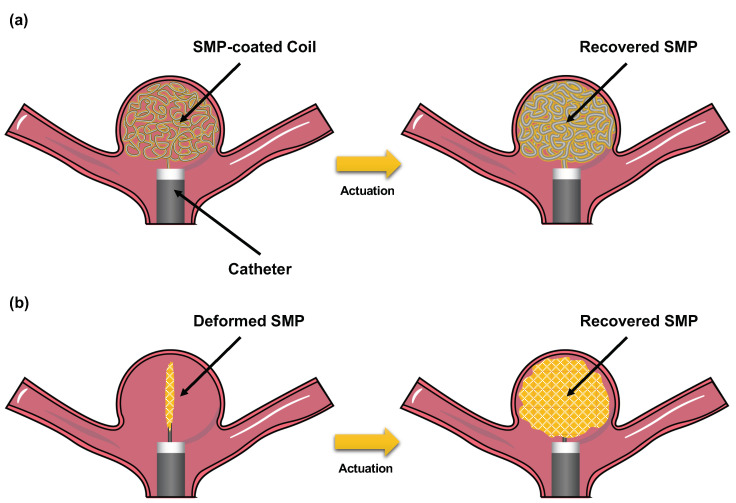
Schematic of two potential approaches to fabricate shape memory polymer (SMP)-based endovascular devices for ICA treatment: (**a**) SMP-coated coils; and (**b**) SMP coil-free foams with patient-specific geometries.

**Figure 3 polymers-14-02526-f003:**
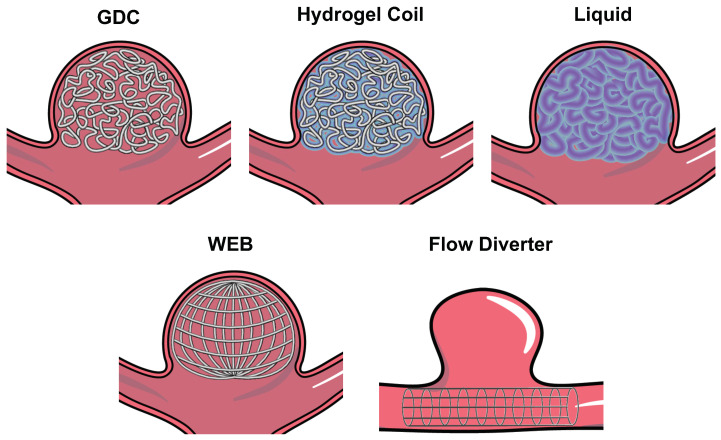
Schematic of different state-of-the-art endovascular devices for occluding (i) “bifurcation” aneurysms (e.g., GDCs, hydrogel coils, liquid embolic devices, and woven EndoBridge—WEB); and (ii) “side-walled” aneurysms (e.g., flow diverter).

**Figure 4 polymers-14-02526-f004:**
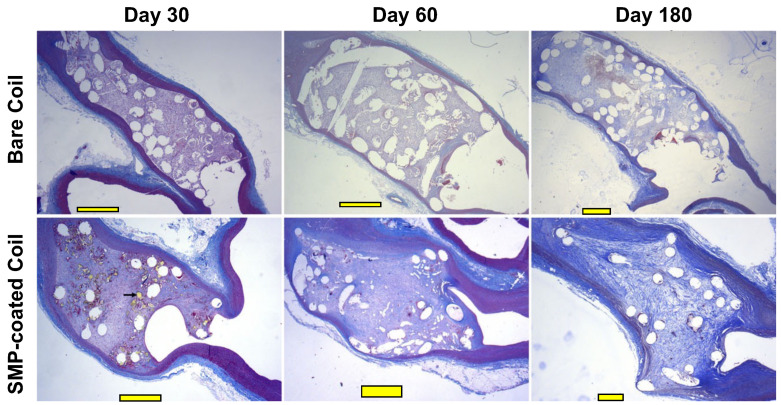
Histological analysis of embolized aneurysms with bare coils vs. SMP-coated coils in a rabbit model. Staining was performed using Masson’s Trichrome at different timepoints (30, 60 and 180 days post-implantation) in this pre-clinical study. Scale bars=1 mm. (Images were adapted from Herting et al. [19] with permission of John Wiley & Sons).

**Figure 5 polymers-14-02526-f005:**
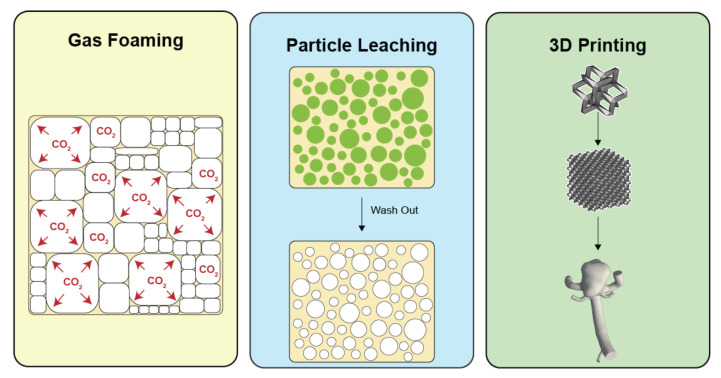
Methods for the fabrication of porous SMPs: CO${}_{2}$ gas foaming (**left**), particle leaching (**middle**), and 3D printing (**right**).

**Figure 6 polymers-14-02526-f006:**
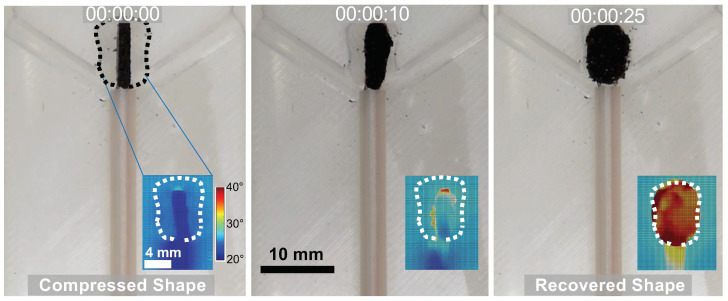
Photos and thermography images (insets) of the shape recovery process of a CNT-infiltrated polyurethane SMP foam developed in our lab. This experiment was performed in an in-vitro bifurcation aneurysm model that mimics a patient-specific aneurysm geometry.

**Table 1 polymers-14-02526-t001:** Current endovascular devices for the treatment of ICAs and their advantages and limitations.

Device	Advantages	Limitations/Complications
GDCs (bare coils)	Gold standard for ICA endovascular treatment	Complete occlusion of 49–65%, high recurrence (44% in 5–6 years) and assisting devices required for the treatment of wide-necked ICAs
Hydrogel coils	Higher packing for improved aneurysm sac filling	Recurrence and complete occlusion are not better than GDCs
Liquid embolic devices	Potential higher packing and maximized aneurysm sac filling	High potential for device migration and FDA approval not granted yet
WEB	Designed for wide-necked aneurysms without assisting device	Limited occlusion, prone to compression in large ICAs, and very high recurrence (∼72%)
Flow diverter	Can treat complex ICA geometry and ideal for side-walled ICAs	Limited geometries can be treated, high delayed rupture and prone to immediate failure
TrelliX	Improved packing density	Not approved by the FDA yet (in clinical trial—NCT03988062)

**Table 2 polymers-14-02526-t002:** Summary of materials with shape memory properties, their activation methods and the most outstanding reported features.

SMP Material	Tested Activation Method	Features	Citation
Butadiene–styrene tri-block copolymer/PCL	Heat/water	Polymer blend with elastomeric and shape memory properties.	[96]
Norland Optical Adhesive 63 (propietary)	Heat	$T_{g}$ gradients have been reported.	[97]
Silsesquioxane nanoparticles/PLA	Heat	Long-term (>1 year) shape storage.	[98]
PCL-coated Fe${}_{3}$O${}_{4}$ /oligo(PCL) DMA/BA	Electromagnetism	Remote activation without reaching temperatures higher than $T_{g}$.	[90]
Perfluorosulphonic acid ionomer	Heat	Multi-shape memory effect with distinctive thermal states.	[99]
Phenylboronic-acid-grafted alginate/PVA	pH	Stable shape storage at specific pH and sugar contents.	[92]
PU/carbon nanopowder	Joule heating	Carbon powder provides conductivity for electrical actuation.	[100,101]
Poly(propylene sebacate)	Heat	Ecologically sustainable material.	[102]
Polydopamine/PCL	Heat	Great thermal conductivity.	[103]
PLA/PPy	Joule heating	Polypyrrole provides conductivity for electrical triggering of shape recovery. PLA is biodegradable.	[51]
Olylactide-co-poly(glycolide-co-caprolactone)	Heat	Highly biodegradable.	[104]
Polymethyl metracrylate/PEG	Heat	Multi-shape memory effect with distinctive thermal states.	[105]
Polystyrene	Heat/IR light	IR activation provides a contactless actuation method (less invasive). Material can also be applied in catheter design. Self-healing properties.	[89,106]
Polystyrene/carbonanotubes	Electromagnetism	Microwave activation provides a minimally invasive actuation method.	[91]
Polystyrene/copolyester particulates	Heat	Self-healing properties.	[107]
PU	Moisture/pH/IR Light	Extensively applied for in-vitro and in-vivo occlusion of aneurysms. Furthermore, used in stent design. Facile control of $T_{g}$ and mechanical properties. Can be designed to be responsive to moisture, IR light, pH, and heat. Multi-shape memory effect with distinctive thermal states can also be achieved.	[55,81,94,95,108,109,110,111]
PU/CNT	Heat/Joule heating	CNTs provide conductivity for electrical triggering of shape recovery.	[58,85]
PU/PEG	Heat	Multi-shape memory effect. Facile $T_{g}$ tuning.	[112]
PU/PPy	Joule heating	Polypyrrole provides conductivity for electrical triggering of shape recovery. PLA is biodegradable.	[83,87,88]
Silver nanowire/PET	Heat	Reported for the design of light-emitting diodes.	[113]
SiO${}_{2}$/PCL	Heat	SiO${}_{2}$ macroparticles provide improved mechanical properties (high strain). Biodegradable.	[114]
Tert-BA/di(ethylene glycol) diacrylate	Heat	It can be fabricated with stereolithography techniques.	[115]
β-CDAlg/DETA-Alg	pH	Stable shape storage at specific pH. Biodegradable and biocompatible.	[93]

## Data Availability

Not applicable.

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
