# Peer review of "Shape Memory Polymer-Based Endovascular Devices: Design Criteria and Future Perspective"

_polymers, 2022, doi:10.3390/polym14132526_

Round 1

Reviewer 1 Report

Authors showed and described about a variety of endovascular embolization devices. In particular, this report focused on recent research trends of shape memory polymers.

If more explanations and data are satisfied in this paper, it will be a good paper.

-      Authors have to make a table for recent research trends of shape memory polymers.

-      Addition of figure or diagram about endovascular embolization devices made by shape memory polymers.

-      It is necessary to explains figures and histochemical staining analysis comparing SMP devices and general endovascular embolization devices for references.

Reviewer 2 Report

Dear authors, congratulations for work. The paper is well written. I suggest you to ampliate the introduction part , giving more informations about what is actually published in literature about this topic. For the same reason, I suggest to read and add to your references two paper which introduced some important concepts about new polymers. The first is: “Non-surgical periodontal treatment of peri-implant diseases with the adjunctive use of diode laser : preliminary clinical study” published in 2016. The other one is : The step further smile virtual planning: milled versus prototyped mock-ups for the evaluation of the designed smile characteristics” published in 2020. 

Thank you for your work.

Author Response

Please see our response in the attached PDF. Thank you!

Reviewer 3 Report

In this article, you systematically illustrate the problems and limitations of current endovascular embolization devices for the treatment of intracranial aneurysms. Shape memory polymers are used in this field because of their versatile shape memory properties. The manuscript provides an overview of the properties of existing SMPs subsequently indicate the advantages and limitations of clinical applications. overall, I think this manuscript can be accepted after minor revisions. It would be even better if you could add more schematics and diagrams to display content visually.  

Author Response

Please see our response in the uploaded PDF. Thank you!

Round 2

Reviewer 1 Report

This study is a focus on the research trends of endovascular devices. In particular, the recent research trend about shape memory polymer was well explained and review. It is believed that it will be good data for endovascular devices research investigators in the future.